# Investigation of the macroscopic behaviour of laminates shells (MBLS) under varying loads using low order CSFE-sh FEM and the N-T's 2-D shell equations

**Joseph Nkongho Anyi**[1,2,3]*, **Alexandra Tchamdjie Pouakam**[1], **Jean Chills Amba**[1], **Fongho Eric**[1], **Platon Dongmo Nizegha**[1], **Merlin Bodol Momha**[1], **Landry Djopkop**[1], **Robert Nzengwa**[1,3]

**1** Laboratoire E3M, Ecole Nationale Supérieure Polytechnique de Douala, Université de Douala, Douala, Cameroun, **2** Department of Mechanical Engineering, Higher Technical Teachers Training College of the University of Buea in Kumba, Kumba, Cameroon, **3** Department of Mechanical Engineering, National advanced School Polytechnics of the University of Yaoundé I–Cameroon, Yaoundé, Cameroon

* nkongho.anyi@ubuea.cm, nkonghojoseph@gmail.com

**Data Availability Statement:** All relevant data are within the paper and its Supporting Information files.

## Abstract

The target in this survey is to investigate deformations of laminates shells (DLS), due to asymmetric and axisymmetric loads, including several other loadings using N-T shell equations. We point out here, the contribution of the metric change in thickness for the analysis of static and linear behavior of laminated composite shells. To achieve this objective, we've applied N-T's shells equations on the same monolayer laminate composite shell and derive the law of MBLS. The macrostructure is analyzed under static loads and implemented using low order curved shell finite elements with shifted Lagrange (CSFE-sh). We tested this element on benchmarks found within the literature. The analysis of cylindrical and spherical shells subjected to uniform sinusoidal pressures and asymmetric pressures reveals excellent accuracy compared to others. The results found without any correction factor were compared with those obtained by the analytical method and other finite element models.

## 1 Introduction

By definition, a composite material is a solid material formed by the association on a microscopic scale of several other materials with complementary characteristics [1]. Two factors are at the origin of the particular behavior of composite materials. The first factor results from the anisotropic behavior of the materials used in the elementary layers, thus making composite materials matter of low shear stiffness. The second factor derives from the stratification, which favors the shear effect of the transverse shear [2].

### 1.1. Structural composite materials

- Monolayers

**Funding:** NO. The funders had no role in study design, data collection and analysis, decision to publish, or preparation of the manuscript.

**Competing interests:** The authors have declared that no competing interests exist.

**Fig 1. Single-layer composite materials** [3].

Monolayers are the basic building block of the composite structure. Types of monolayers are characterized by the shape of the reinforcement which can be long fibers (unidirectional (UD) and randomly distributed), woven fibers, short fibers. In a UD sheet, the fibers are assembled parallel to each other using a very light weft (Fig 1), and the rate of imbalance is very large.

• Laminates

  Laminates are successive layers (sometimes called plies) of reinforcements (threads, roving, mats, fabrics, etc.) impregnated with resins. There are types of laminates, some are frequently used like Laminates based on unidirectional yarns or fabrics are consist of a stack of monolayers (Fig 2) adhering to one another, each having its orientation concerning a frame of reference common to the layers and designated as the frame of the laminate.

• Sandwiches

  Materials made of two soles (or skins) of great rigidity and of low thickness envelop a core (or core) of great thickness and low resistance (Fig 3). The sandwich material has a great lightness in bending it is an excellent thermal insulator. The most important application sectors are therefore naval, space, aeronautics and sport.

## 1.2. Models or theories of composite shells

A plethora of theories and formulations for the modeling and analysis of composite plates and shells adapted to finite element modeling, make it possible to reduce the three-dimensional problem to a two-dimensional problem and to respect the conditions resulting from the mechanics of the media Continues have been developed in recent decades. In the literature, there are several theories taking into account the effect of transverse shear for the analysis of

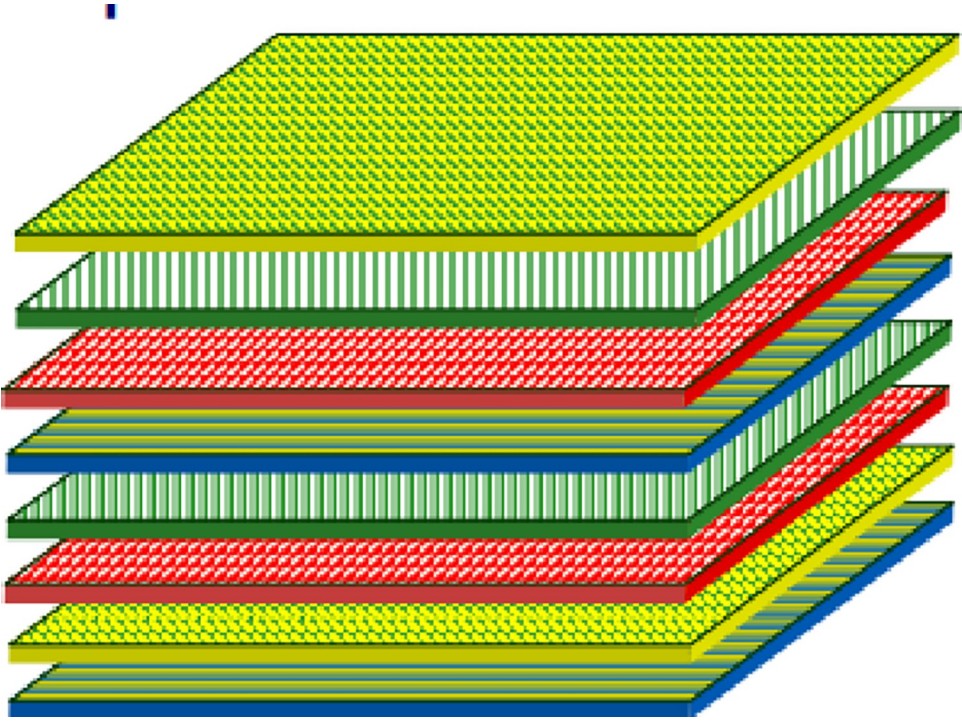

**Fig 2. Laminate composite materials.**

multilayer composite structures. Let us quote for example [4, 5] which contains the different models and theories. The three-dimensional 3D theories which are often used in the analysis of multi-layered composite structures allow very precise and exact results to be obtained, but they are limited to some simple examples of geometry, stacking, and loading. Y. M. Ghugal and R. P. Shimpi [6] have devoted a section to exact three-dimensional (3D) elasticity methods, such as those developed by Pagano, Srinivas, and Rao and who have proposed analytical solutions to some simple cases of problems. These solutions are considered to be references in the literature for the analysis of isotropic and composite laminates [7]. Furthermore, these approaches can lead to differential equations that cannot be solved analytically. Therefore,

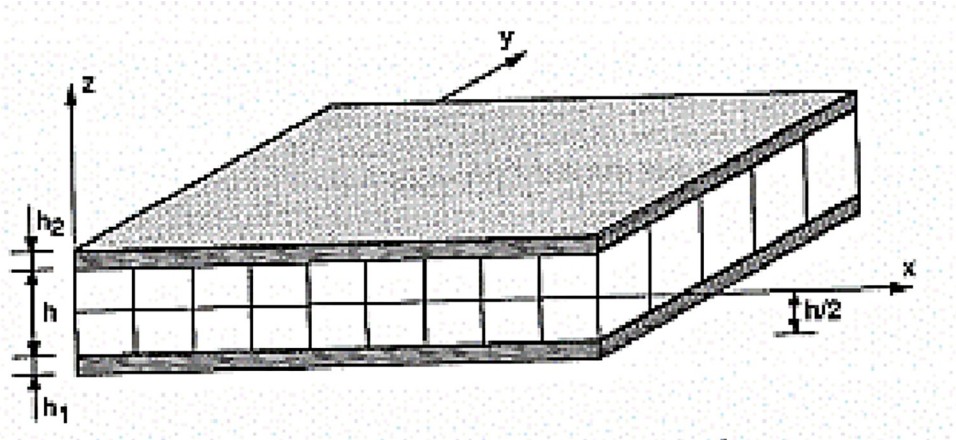

**Fig 3. Sandwich plate.**

mathematical models and numerical methods such as the finite element method have been proposed for the modeling of multilayer structures.

- Equivalent monolayer theories, by layer

These include the equivalent single-layer (ESL) and the layer-based approaches (LW). These two theories depend on the displacement model with a thematic description derives from the kinematics of multilayer plates/shells, based on the assumption of variation of the displacement field through the thickness. They are into two general classes (Fig 4): Equivalent Single Layer Theories (ESLM) and Layer Theories (Layer-Wise).

Among the different classes of theories, the ESL approach is the most frequently used. It is on the idea of representing a complex heterogeneous laminate by a single statically equivalent lamina (homogenization), which reduces problems of complex 3D elasticity to a much simpler 2D problem. The major disadvantage of ESLM models in modeling multilayer composites is that the shear strain components are continuous across the interfaces of different material layers. Thus, the transverse stress components are discontinuous at the interfaces of the layers. This drawback appears in relatively thick multilayer structures, or in local regions subject to complex loads or to geometric or material discontinuities [7]. These successful models which neglect the transverse normal strain are not able to correctly determine the inter-laminar stresses close to discontinuities such as holes [7]. Two reasons show the importance of including the transverse normal strain in the modeling of these local effects: first, the transverse normal stress is generally significant stress in these regions, if it is not the dominant one. Second, as reported by Robbins and Reddy [8], layered models which neglect transverse normal strain do not satisfy the tensile boundary conditions for transverse shear stresses at the free edges of the laminate composite. An examination of the natural boundary conditions for the differential equations of motion developed in references [9–11] (for the case of layered theories which neglect the transverse normal strain) reveals that the transverse shear stresses satisfy the boundary conditions in tension at the free edge of the structure, in the integral sense only and not in the local sense (in spite of the level of refinement according to the thickness).

- Global theories of deformations

During recent decades the modelling approaches, which require consideration of the transverse shear strain, have been the subject of serious research. Some modelling approaches are extensions of some similar to those used in isotropic plates/shells. Those belonging to the category called "Smeared Laminate Models" are models based on global assumptions (for the entire stratification). They are characterized by a linear or non-linear distribution of plane displacements or transverse shear stresses according to the direction of the thickness. These approaches consider all the layers as an equivalent anisotropic monolayer [12–14]. Two theories, widely discussed in the specialized literature, are based on a global linear distribution.

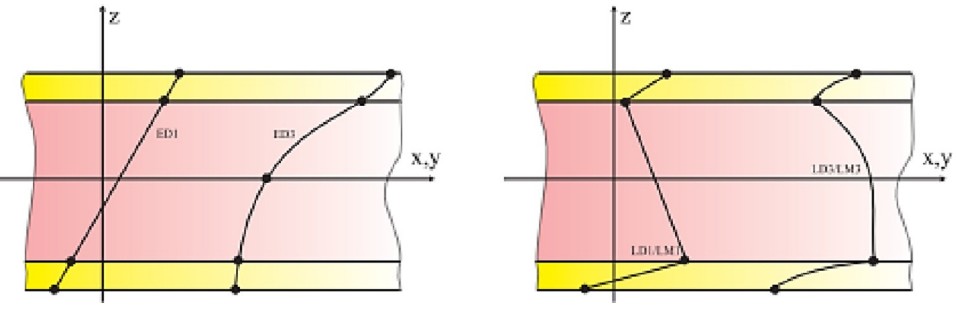

**Fig 4. Example of the ESL (left) and LW (right) hypothesis [6].**

## 1.3. Recent works on finite element models of behavior modeling of stratified structures

During recent decades solid shell EF models for the thin structure have been studied considerably. An ideal shell element for composite structures should enable the modeling of curved shapes of a shell of arbitrary geometry. The formulation of this element should be simple, avoiding as much as possible derived shifts such as nodal variables, which require higher-order continuities. The main challenge for low degrees of Lagrange shell elements, based on the Naghdi-Reissner-Mindlin hypotheses, concerns membrane and shear blocking [15].

Several authors have worked on the properties of solid shell elements suitable for modeling layered structures. Cinefra Maria and Carrera Erasmo [16] analyzed multilayer structures under thermal and electromechanical loading. They used the finite element shell MITC9 based on Carrera's unified formulation. A Nine-node shell element and MITC method are used to contrast the phenomenon of membrane and shear locking. Results obtained through this approach compared to those obtained analytically using the unified Carrera approach and the Navier method. These turn out to be satisfactory. H. Naceur, S. Shiri, D. Coutellier, J.L. Batoz [17] have proposed a finite element formulation of a developed 8-node composite element called SCH8γ7 based on only one translational degree of freedom. The anisotropic behavior of layered shells was derived from a fully three-dimensional elastic orthotropic material's law in each layer, including the thickness stresses components. Irwan Katili, Imam Jauhari Maknun, Jean-Louis Batoz, Adnan Ibrahimbegovic [18] propose a new 4-node DKMQ24 hull element based on the Naghdi-Reissner-Mindlin hull theory with 24 degrees of freedom. This new composite shell element, developed from DKMQ plate and shell elements, consider the shear strain, the coupled energy of the bending membrane, and the effects of warping. Numerical results obtained from DKMQ24, compared to state-of-the-art shell elements, converge more quickly to the reference solution. Irwan Katili, Imam Jauhari Maknun, Jean-Louis Batoz, Adnan Ibrahimbegovic [19] proposed an efficient 3-node shell element with 6 degrees of freedom based on Naghdi-Reissner-Mindlin theory. This new composite shell element, also called DKT18, considers the shear strain and the coupled energy of the bending membrane. This shell element is capable of calculating composite layered structures and does not present any shear blockage problem.

In the literature, there's a good number of works on the studies of deformations of composite shells. But very little based on higher-order theories. Most of the theoretical results are illustrated by finite element numerical calculations using adaptive and anisotropic meshing techniques. These solve the cost problem encountered when using the refined meshes, which tend to be isotropic throughout the shell, thus including very high costs and very long resolution times. Ajay Kumar, Anupam Chakrabarti and Mrunal Ketkar [20] performed a static analysis of asymmetric composite shells using a Co finite element model based on the theory of higher-order shear strain (HSDT). In this theory, transverse shear stresses referenced at zero at the top and the bottom of the shell. It assumes a parabolic variation in transverse shear strains where the correction factor hasn't been of use. The assumption on Sander's approximations includes the three curvature terms in the strain components of composite shells. The finite element used is isoperimetric of 9 nodes with 7 unknowns per node. The lack of work or results available in the literature on HSDT-based asymmetric shell problems prompted them to validate the results with the limited work. Mantari, Guedes Soares, Oktem [21] perform static and dynamic analysis of laminated composite plates and shells using a new theory of higher-order shear strain. It takes into account the parabolic distribution of the transverse shear strains through the thickness, and therefore no longer considers the correction factor. The thin composite shell is, therefore, subjected to distributed and point bi-sinusoidal loads.

Erasmo Viola, Francesco Tornabene and Nicholas Fantuzzi [22] perform static analysis of double-curvature laminated composite shells and plates. It offers a 2D theoretical formulation of higher-order deformation theory based on the equivalent single-layer approach. With the differential geometry, the mid surface of the shells and plates is described. The numerical problem is solved using the technique of generalized differential quadrature (GDQ). Results obtained by this technique, compared with those obtained in the literature with semi-analytical methods and those calculated by the finite element method. Francesco Tornabene and Michele Bacciocchi [23] propose a weak higher-order formulation for double-curvature layered composite shell structures of arbitrary shape. The theoretical shell model lays on the equivalent single-layer approach. The numerical tool used to guarantee a high level of precision with a low computational effort is the generalized integral quadrature (GIQ) technique.

As already discussed above and taking into account the global or local character used in the description of the field of displacement (or of stress), one can distinguish various approaches generally based on the theory of Kirchhoff for the structures thin or on the Reissner-Mindlin theory for thick shell structures. We classify them either as global approaches (whose shell structure is considered equivalent homogeneous) or as local (or combined global-local) called "by layer" in which the displacement or stress field is dependent on a layer and varies linearly or more depending on the thickness. If considered this numerical performance-linked proportionately to the high cost of computation means, these latter approaches are generally of good use for specific applications for which the local response is required and/or when the local behavior is dominant in certain places. Nevertheless, the classical model remains more widespread in the modeling of geometrically complex composite shell structures, it's a reason to correct the components of the stiffness in shearing is sometimes necessary to improve the behavior with the shearing force. The obtained results depends essentially on the choice of the correction coefficients and the estimate of the stresses in some thick structures is uncertain and requires a careful model. To overcome this anomaly while keeping the operational character of this equivalent homogeneous model, one proposes like a goal, to extend the N-T model [24–26] which reflects a displacement field by layer, on a finite element model for multilayer shell structures. The finite element formulation for the modeling of composite shells will be presented in the rest of the work. The elements are curved triangles with three nodes, based on the concept of N-T, they take into account the multi-layered aspect of the structure. To achieve our target, a simple mathematical description of a multi-layer shell will be made first, next a wick formulation of laminated domain energy will be presented using strain tensor found in N-T equations. Further, CSFE-sh is going to be used to discretize the energy balance equation leading to a non-classical stiffness matrix. Finally, the obtained unusual equation of MBLS is tested of standard benchmarks of composite laminated shells.

## 2. Materials and methods

### 2.1. Thin composite shells

Thin shell structures are the most used, especially in engineering, because they have many advantages such as; high rigidity, good strength, efficient behavior under load, space containment. Apart from these mechanical properties, in some architectural designs shell structures have an aesthetic appearance. Volumetric composites have been used to solve specific problems encountered in aeronautics. Despite their very high cost and in addition to their specificities, they make it possible to obtain high mechanical characteristics, with a substantially isotropic behavior in volume [22]. According to theory of [17], composite shells are structures composed of several layers (Fig 5) with struts of finite rigidity bendable in the longitudinal direction and rigid in the transverse direction.

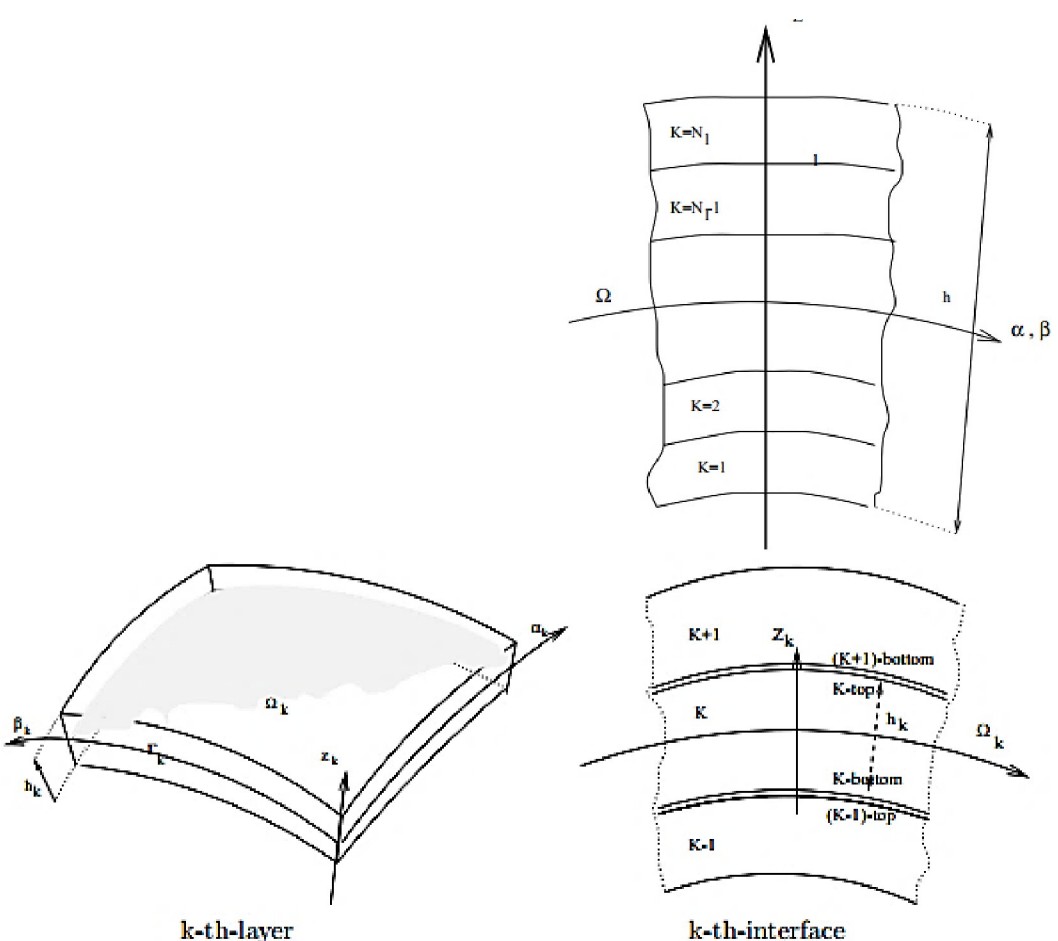

**Fig 5. Multilayer shell (Carrera 2003).**

• 3D geometric description of a shell

A shell is a solid bounded by two nearby and approximately parallel surfaces. It must be closed on itself, or delimited in addition by a peripheral surface (the edge) which joins the two main surfaces. A complete shell is a three-dimensional solid bounded by two curved surfaces separated by a distance called the thickness of the shell. Let M be a point on the shell, P a point on the mid surface S and the curvilinear coordinates $(x^1, x^2, x^3)$ with $x^3 = z$. A complete shell can be defined as follows:

$$\Omega = \left\{ \begin{array}{c} M \in \mathbb{R}^3, \boldsymbol{OM} = \boldsymbol{\Phi}(x^1, x^2, x^3) = \boldsymbol{OP}(x^1, x^2) + z\boldsymbol{a}_3(x^1, x^2), \boldsymbol{OP}(x^1, x^2) \in S, \\ -\dfrac{h}{2} \leq z \leq \dfrac{h}{2} \end{array} \right\}$$

## 2.2. Development of the model of macroscopic behavior of the laminate shells

For a rigorous modeling, the following notations must be made. Each ply k is considered thin. Thanks to the following Fig 6, we give the kinematics of a bend.

$\Omega$ is the domain of the laminate, $\Omega_k$ the domain of the ply k and $e_k$ is the thickness of a ply k. S denotes the mid surface of the laminate and $S_k$ the mid surface of a ply k which is a $\mathbb{R}^3$ domain. $n_{i,k-1}^k$ is the normal unit vector directed from ply k-1 to ply k and $n_{i,k}^{k-1}$ is the normal

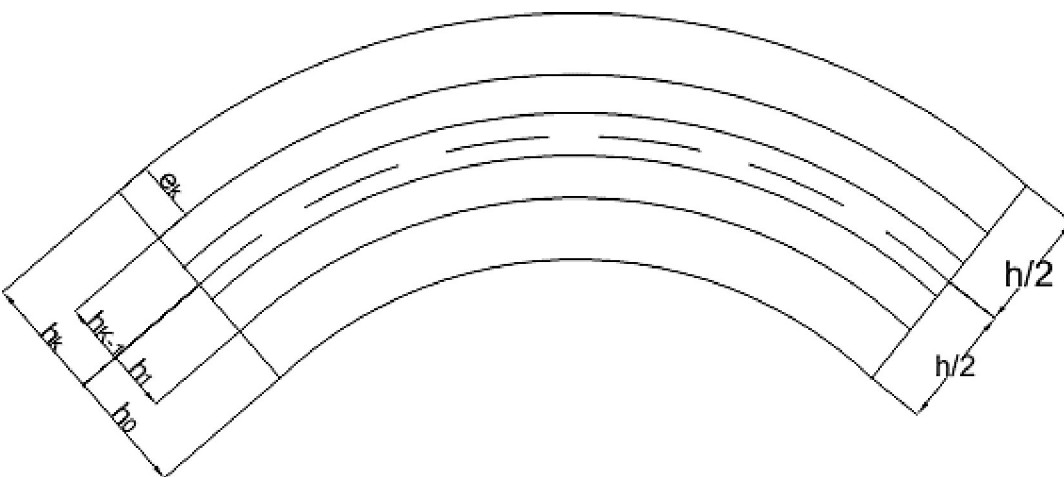

**Fig 6. Layer stacking.**

unit vector directed from the ply k to the ply k-1. $\Gamma_k^-$ is the lower edge of ply k and $\Gamma_k^+$ is the upper edge of the ply k. $\Gamma_{k-1,k}$ represents the interface between two plies k-1 and k. $\partial\Omega$ is the boundary of the laminate. $\partial\Omega_k$ is the boundary of ply k.

A point P located on the mean surface $S_k$, a point M located in a fold k is defined by:

$$\Omega_k = \left\{ \begin{array}{c} M \in \mathbb{R}^3, OM = \Phi(x^1, x^2, x^3) = OP(x^1, x^2) + za_3(x^1, x^2), OP(x^1, x^2) \in S, \\ h_{k-1} \le z \le h_k \end{array} \right\} \tag{2.1}$$

$$\partial\Omega_k = \Gamma_k^- \cup \Gamma_k^+ \times \{h_{k-1}, h_k\} \tag{2.2}$$

- General assumptions

H1: All the points on a normal to the mean plane of the shell have the same transverse displacement.

H2: It is assumed that the layers of the composite laminate are perfectly bonded together so that the displacement field is continuous between the laminated plies.

H3: the constituent materials of the shell are assumed to be homogeneous and elastic.

H4: the Poisson's ratio and the different moduli of materials do not change over time.

H5: each ply is considered thin enough that we can assume it in a plane strain state.

H6: the normal unit vector to a ply coincides with that of the laminate taken in its structure ($\overrightarrow{n}_k = \overrightarrow{n}$).

H7: assumptions of inter-laminar stresses: The plane stresses can be discontinuous in the ply but the inter-laminar continuities conditions of the transverse normal stresses are imposed.

$((\sigma_{k-1}n_{i,k-1}^k = \sigma_k n_{i,k}^{k-1}.)$. This assumption is of great importance in any suitable development of multi-ply composite laminates.

H8: continuity of displacements at the interfaces: $u^{k-1}_{\Gamma_{k-1,k}} = u^{k}_{\Gamma_{k-1,k}}$. No-consideration of the phenomenon of delamination of the laminate.

H9: the normal stress in the thickness is assumed to be non-zero.

H10: the curvature of the shell is assumed to be uniform; each layer is assumed to be of uniform thickness.

H11: one supposes the stresses and strains are small.

H12: the general assumptions of the composite shells in particular those of the laminated shells are applied. The transverse shear stresses are not all zero ($\sigma_{iz} \neq 0$, $i = x,y$).

## 2.3 Modeling of the macroscopic behavior of the laminate

Macro-mechanical property survey of a three-dimensional linear elastic behavior of a ply of a laminate composite shell consists of knowing the stress and strain field of the layer (or ply) k considered to be a thin shell. For a laminate shell structure embedded at its ends, we have:

$$\begin{cases} -div\,[\sigma]_k = f \ in \ \Omega_k \\ [\sigma].n = P \ on \ \partial\Omega_k \\ (\sigma_{k-1} n^{k}_{i,k-1} = \sigma_k n^{k-1}_{i,k} \ sur \ \Gamma_{k-1,k} \ continuity \ of \ stresses \\ u^{k-1}_{\Gamma_{k-1,k}} = u^{k}_{\Gamma_{k-1,k}} \ sur \ \Gamma_{k-1,k} \ continuity \ of \ displacements \\ u_i = \theta_i = 0 \ sur \ le \ bord \ (c.a.l) \ \ u_i = (u,v,w) \end{cases} \tag{2.3}$$

- Law of behavior of a laminated shell

Each shell layer is assumed to be a linear elastic material exhibiting monoclinic symmetry. The components $Q^{ijkl}$ of Q in the covariant basis of space (at point M) satisfy the symmetry properties $Q^{ijkl} = Q^{ijlk} = Q^{klij} = Q^{jikl}$. From above, Hooke's law on a ply k reads:

$$\sigma^k = [Q]_k \varepsilon^k. \tag{2.4}$$

One can expressed the Hooke's law on a ply k using matrix as below:

$$\begin{pmatrix} \sigma_{11} \\ \sigma_{22} \\ \sigma_{33} \\ \sigma_{23} \\ \sigma_{31} \\ \sigma_{12} \end{pmatrix}_k = \begin{pmatrix} Q_{11} & Q_{12} & Q_{13} & 0 & 0 & 0 \\ Q_{21} & Q_{22} & Q_{23} & 0 & 0 & 0 \\ Q_{31} & Q_{33} & Q_{33} & 0 & 0 & 0 \\ 0 & 0 & 0 & Q_{44} & 0 & 0 \\ 0 & 0 & 0 & 0 & Q_{55} & 0 \\ 0 & 0 & 0 & 0 & 0 & Q_{66} \end{pmatrix}_k \begin{pmatrix} \varepsilon_{11} \\ \varepsilon_{22} \\ \varepsilon_{33} \\ 2\varepsilon_{23} \\ 2\varepsilon_{31} \\ 2\varepsilon_{12} \end{pmatrix}_k. \tag{2.5}$$

The bidimensional strain tensor of the ply k reads:

$$\varepsilon(U) = \varepsilon_{\alpha\beta}(U) x_\alpha \otimes x_\beta, \tag{2.6}$$

where

$$\varepsilon = \varepsilon_{\alpha\beta}(U) = e_{\alpha\beta}(\tilde{u}) - z k_{\alpha\beta}(\tilde{u}) + z^2 Q_{\alpha\beta}(\tilde{u}), \tag{2.7}$$

see [25].

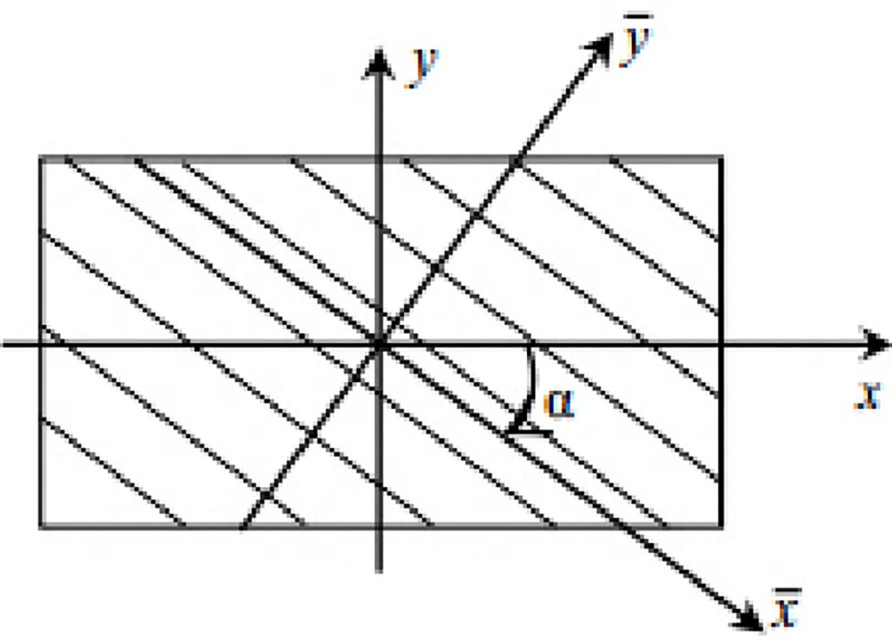

**Fig 7. Orthotropy basis and local ply basis.**

- Basis change criterion

In the case of multi-layered composite materials with different fiber orientations, the system coordinates are different from those of the material. It is, therefore, necessary to make a change of base (Fig 7) to find the stiffness matrix. Let $[\bar{Q}]$ be a stiffness matrix defined in the orthotropy basis, $[Q]$ a stiffness matrix defined in a basis that differs from the orthotropy axis. These two matrices are connected as follow:

$$[Q] = T_\sigma^{-1}[\bar{Q}]T_\varepsilon; \tag{2.8}$$

with $T_\sigma$ such that;

$$T_\varepsilon = (T_\sigma^{-1})^t; \tag{2.9}$$

$$T_\sigma = \begin{bmatrix} cos^2\alpha & sin^2\alpha & 0 & 0 & 0 & 2cos\alpha sin\alpha \\ sin^2\alpha & cos^2\alpha & 0 & 0 & 0 & -2cos\alpha sin\alpha \\ 0 & 0 & 1 & 0 & 0 & 0 \\ 0 & 0 & 0 & cos\alpha & -sin\alpha & 0 \\ 0 & 0 & 0 & sin\alpha & cos\alpha & 0 \\ -cos\alpha sin\alpha & cos\alpha sin\alpha & 0 & 0 & 0 & cos^2\alpha - sin^2\alpha \end{bmatrix}; \tag{2.10}$$

where $\alpha$ is the angle between the reference ply and the orthotropy basis in which

$$\sigma' = T_\sigma \sigma; \tag{2.10.A}$$

$$\varepsilon' = T_\sigma \varepsilon. \tag{2.10.B}$$

$\varepsilon'$ denote the strain in the orthotropy basis and $\varepsilon$ the strain in a basis out of orthotropy axis.

$\sigma'$ denote the stress in the orthotropy basis and $\sigma$ the stress in a basis out of orthotropy axis.

- Homogeneous stiffness Matrix of laminated composites

  It's obtained by inversing the flexibility matrix as below

$$S^{-1} = \lfloor Q \rfloor = \begin{pmatrix} Q_{11} & Q_{12} & Q_{13} & 0 & 0 & 0 \\ Q_{21} & Q_{22} & Q_{23} & 0 & 0 & 0 \\ Q_{31} & Q_{33} & Q_{33} & 0 & 0 & 0 \\ 0 & 0 & 0 & Q_{44} & 0 & 0 \\ 0 & 0 & 0 & 0 & Q_{55} & 0 \\ 0 & 0 & 0 & 0 & 0 & Q_{66} \end{pmatrix} = \begin{pmatrix} Q_1 & Q_2 \\ Q_3 & Q_4 \end{pmatrix}; \qquad (2.11)$$

$$\lfloor Q_1 \rfloor = \begin{pmatrix} \dfrac{1}{\Delta E_2 E_3}(1 - v_{23}v_{32}) & \dfrac{1}{\Delta E_2 E_3}(v_{21} + v_{31}v_{23}) & \dfrac{1}{\Delta E_2 E_3}(v_{21}v_{32} + v_{31}) \\ \dfrac{1}{\Delta E_1 E_3}(v_{12} + v_{13}v_{32}) & \dfrac{1}{\Delta E_1 E_3}(1 - v_{13}v_{31}) & \dfrac{1}{\Delta E_1 E_3}(v_{32} + v_{12}v_{31}) \\ \dfrac{1}{\Delta E_1 E_2}(v_{12}v_{23} + v_{13}) & \dfrac{1}{\Delta E_1 E_2}(v_{23} + v_{13}v_{21}) & \dfrac{1}{\Delta E_1 E_2}(1 - v_{12}v_{21}) \end{pmatrix}; \qquad (2.12)$$

with $\qquad \Delta = \dfrac{1 - v_{23}v_{32} - v_{31}v_{13} - v_{12}v_{21} - 2v_{21}v_{13}v_{32}}{E_1 E_2 E_3}.$ $\qquad (2.13)$

Broadly speaking, for $i,j,k = \langle 1,2,3 \rangle$,

$$Q_{ij} = \begin{cases} \dfrac{1}{\Delta E_1 E_j}\left(1 - v_{ij}v_{ji}\right) & si\ i = j \\[2ex] \dfrac{1}{\Delta E_i E_j}\left(v_{ik} + v_{ij}v_{ji}\right) & si\ i \neq j \end{cases}; \qquad (2.14)$$

$$\lfloor Q_2 \rfloor = \lfloor Q_3 \rfloor = 0; \qquad (2.15)$$

$$\lfloor Q_4 \rfloor = \begin{pmatrix} G_{23} & 0 & 0 \\ 0 & G_{13} & 0 \\ 0 & 0 & G_{12} \end{pmatrix}. \qquad (2.16)$$

## 2.4 Mathematical modeling of the behavior of the laminate

- Variational formulation

The reference elasticity problem (P) is defined by the set of Eqs (2.17) and (2.18). Let $U_{ad}$ be the fields of kinematically admissible displacements, i.e., that satisfies the boundary conditions in displacements and $\Sigma_{ad}$ the fields of the statically admissible stresses, i.e., that satisfies the equilibrium equations and the boundary conditions in efforts. If $U$ is a field of displacements,

the solution of problem (P) then yields:

$$\begin{cases} U \in U_{ad} \\ a(U, \mathrm{V}) = L(\mathrm{V}) \qquad \forall \mathrm{V} \in V_{ad} \end{cases}. \tag{2.17}$$

From the following equation that consists:

$$\begin{cases} \text{to find } \mathrm{U} \in IH_{\Gamma_0}^1 \\ \int_\Omega - div\, \sigma_k(U).V\mathrm{d}\Omega_k = \int_\Omega f^i.V_i\, d\Omega, \ \ \forall V \in V_{ad} \end{cases}, \tag{2.18}$$

$IH_{\Gamma_0}^1(\Omega) = \{u_i : \Omega \to \mathbb{R}, \ u_i \in L^2(\Omega); \nabla_j u_i \in L^2(\Omega) \text{ et } u_i = 0 \text{ sur } \Gamma_0\} = V_{ad}$ is a Sobolev's space; we deduce:

$$\int_\Omega \sigma_k^{\ t}(U).\boldsymbol{\varepsilon}(V)\ d\Omega_k = \int_\Omega f^i.V_i\, d\Omega + \int_\Omega (\sigma(U).n).\nabla V\, dS = L(V), \ \ \forall V \in V_{ad}; \tag{2.19}$$

$$\int_\Omega \sigma_k^{\ t}(U).\boldsymbol{\varepsilon}(V)\mathrm{d}\Omega_k = \int_\Omega f^i.V_i\, d\Omega_k + \int_{\partial\Omega} g^{-i}.V_i\, dS = L(V), \ \ \forall V \in V_{ad}; \tag{2.20}$$

$$\int_\Omega \sigma^{\alpha\beta}(\tilde{u}).\boldsymbol{\varepsilon}_{\alpha\beta}(\tilde{v})\mathrm{d}\Omega_k = \int_\Omega f^i.V_i\, d\Omega_k + \int_{\partial\Omega} g^{-i}.V_i\, dS = L(V), \ \ \forall V \in V_{ad}; \tag{2.21}$$

with $\sigma^T = \{\sigma^{11}, \sigma^{22}, \sigma^{12}, \sigma^{23}, \sigma^{13}, \sigma^{33}\}$ and $\boldsymbol{\varepsilon}^T = \{\boldsymbol{\varepsilon}^{11}, \boldsymbol{\varepsilon}^{22}, \boldsymbol{\varepsilon}^{12}, \boldsymbol{\varepsilon}^{23}, \boldsymbol{\varepsilon}^{13}, \boldsymbol{\varepsilon}^{33}\}$.

The stated problem above simply consists now

$$\begin{cases} \text{to find } \mathrm{U} \in IH_{\Gamma_0}^1 \\ \int_\Omega \sum_{k=1}^n [C]_k \simeq \boldsymbol{\varepsilon}^\sim(U) : \simeq \boldsymbol{\varepsilon}^\sim(V)d\Omega = \int_\Omega f.V\, d\Omega + \int_{\partial\Omega} g.V\, dS = L(V), \\ \qquad\qquad\qquad \forall V \in IH_{\Gamma_0}^1 \end{cases}. \tag{2.22}$$

This is equivalent as

$$\begin{cases} \text{to find } \xi = (\xi_\alpha(x), \xi_3(x)), \xi_i = 0 \text{ et } \partial_\alpha \xi_3 = 0 \text{ sur } \Gamma_k^+ \text{ such that} \\ \int_\Omega \sum_{k=1}^n [C]_k (e_{\alpha\beta}(\xi) - zk_{\alpha\beta}(\xi) + z^2 Q_{\alpha\beta}(\xi)) : (e_{\alpha\beta}(\eta) - zk_{\alpha\beta}(\eta) + z^2 Q_{\alpha\beta}(\eta))\, d\Omega \\ = L(\eta), \ \eta = (\eta_\alpha(x), \eta_3(x)). \end{cases} \tag{2.23}$$

The first-order approximation of $(\mu^{-1})_\rho^\alpha$ for n = 1 and its integration with respect to the thickness, is discussed in [25]. The strain tensor $\boldsymbol{\varepsilon} = [\varepsilon_{ij}]; \varepsilon_{\alpha\beta} = e_{\alpha\beta} - zk_{\alpha\beta} + z^2 Q_{\alpha\beta}$, substituted by its components, membrane strain tensor $e_{\alpha\beta}$, curvature strain tensor $k_{\alpha\beta}$ and Gauss curvature strain tensor $Q_{\alpha\beta}$, heads

$$\begin{cases} \text{to find } \xi = (\xi_\alpha(x), \ \xi_3(x)), \xi_i = 0 \text{ et } \partial_\alpha \xi_3 = 0 \text{ sur } \gamma_0, \\ \int_\Omega [C_k]\ (e^{\rho\gamma}(\xi) - zk^{\rho\gamma}(\xi) + z^2 Q^{\rho\gamma}(\xi)) : (e_{\alpha\beta}(\eta) - zk_{\alpha\beta}(\eta) + z^2 Q_{\alpha\beta}(\eta))d\Omega, \\ = L(\eta), \ \eta = (\eta_\alpha(x), \ \eta_3(x)). \end{cases} \tag{2.24}$$

Let considered $A_{cs}(u,v)$ to be the approximated internal energy of the laminated composite shell

$$A_{cs}(u,v) = \int_s \sum_{k=1}^{n} \int_{h_{k-1}}^{h_k} [C_k] \begin{bmatrix} e_{\alpha\beta}(\eta)e^{\rho\gamma}(\xi) - zk^{\rho\gamma}(\xi)e_{\alpha\beta}(\eta) + z^2 Q^{\rho\gamma}(\xi)e_{\alpha\beta}(\eta) \\ -ze^{\rho\gamma}(\xi)k_{\alpha\beta}(\eta) + z^2 k^{\rho\gamma}(\xi)k_{\alpha\beta}(\eta) - z^3 Q^{\rho\gamma}(\xi)k_{\alpha\beta}(\eta) \\ +z^2 e^{\rho\gamma}(\xi)Q_{\alpha\beta}(\eta) - z^3 k^{\rho\gamma}(\xi)Q_{\alpha\beta}(\eta) + z^4 Q^{\rho\gamma}(\xi)Q_{\alpha\beta}(\eta) \end{bmatrix} ds dz, \quad (2.25A)$$

by computing through the layer thickness, one finds

$$A_{cs}(u,v) = \int_s \sum_{k=1}^{n} [C_k] \begin{pmatrix} (h_{k-1} - h_k)e_{\alpha\beta}(v)e^{\rho\gamma}(u) - \dfrac{(h_{k-1}^2 - h_k^2)}{2} k^{\rho\gamma}(u)e_{\alpha\beta}(v) + \dfrac{(h_{k-1}^3 - h_k^3)}{3} Q^{\rho\gamma}(u)e_{\alpha\beta}(v) \\ -\dfrac{(h_{k-1}^2 - h_k^2)}{2} e^{\rho\gamma}(u)k_{\alpha\beta}(v) + \dfrac{(h_{k-1}^3 - h_k^3)}{3} k^{\rho\gamma}(u)k_{\alpha\beta}(v) - \dfrac{(h_{k-1}^4 - h_k^4)}{4} Q^{\rho\gamma}(u)k_{\alpha\beta}(v) \\ +\dfrac{(h_{k-1}^3 - h_k^3)}{3} e^{\rho\gamma}(u)Q_{\alpha\beta}(v) - \dfrac{(h_{k-1}^4 - h_k^4)}{4} k^{\rho\gamma}(u)Q_{\alpha\beta}(v) + \dfrac{(h_{k-1}^5 - h_k^5)}{5} Q^{\rho\gamma}(u)Q_{\alpha\beta}(v) \end{pmatrix} ds. \quad (2.25B)$$

A simple matrix expression can easily be deduced from Eqs (2.25A) and (2.25B). $A_{cs}(u,v)$ is now red:

$$A_{cs}(u,v) = \int_s [N(u) : e(v) + M(u) : K(v) + M^*(u) : Q(v)] ds = \int_s [N, M, M^*] \begin{bmatrix} e \\ k \\ Q \end{bmatrix} ds = L(v). \quad (2.26)$$

Inner forces and moments components are now identified in Eq (2.26) and computed below

$$[N, M, M^*] = [e^{\rho\gamma}, k^{\rho\gamma}, Q^{\rho\gamma}]$$

$$\times \begin{pmatrix} \sum_{k=1}^{n}[C_k](h_{k-1} - h_k) & -\dfrac{1}{2}\sum_{k=1}^{n}[C_k](h^2_{k-1} - h^2_k) & \dfrac{1}{3}\sum_{k=1}^{n}[C_k](h^3_{k-1} - h^3_k) \\ -\dfrac{1}{2}\sum_{k=1}^{n}[C_k](h^2_{k-1} - h^2_k) & \dfrac{1}{3}\sum_{k=1}^{n}[C_k](h^3_{k-1} - h^3_k) & -\dfrac{1}{4}\sum_{k=1}^{n}[C_k](h^4_{k-1} - h^4_k) \\ \dfrac{1}{3}\sum_{k=1}^{n}[C_k](h^3_{k-1} - h^3_k) & -\dfrac{1}{4}\sum_{k=1}^{n}[C_k](h^4_{k-1} - h^4_k) & \dfrac{1}{5}\sum_{k=1}^{n}[C_k](h^5_{k-1} - h^5_k) \end{pmatrix}. (2.27)$$

It appears from the above-disclosed components of inner forces and moments that for the best first-order approximation additional components are found in the matrix $[A]$ and make it different from the one obtained with classical shell theories,

$$[A] = \begin{pmatrix} \sum_{k=1}^{n}[C_k](h_{k-1} - h_k) & -\dfrac{1}{2}\sum_{k=1}^{n}[C_k](h^2_{k-1} - h^2_k) & \dfrac{1}{3}\sum_{k=1}^{n}[C_k](h^3_{k-1} - h^3_k) \\ -\dfrac{1}{2}\sum_{k=1}^{n}[C_k](h^2_{k-1} - h^2_k) & \dfrac{1}{3}\sum_{k=1}^{n}[C_k](h^3_{k-1} - h^3_k) & -\dfrac{1}{4}\sum_{k=1}^{n}[C_k](h^4_{k-1} - h^4_k) \\ \dfrac{1}{3}\sum_{k=1}^{n}[C_k](h^3_{k-1} - h^3_k) & -\dfrac{1}{4}\sum_{k=1}^{n}[C_k](h^4_{k-1} - h^4_k) & \dfrac{1}{5}\sum_{k=1}^{n}[C_k](h^5_{k-1} - h^5_k) \end{pmatrix}, (2.28)$$

$$[N, M, M^*] = [e^{\rho\gamma}, k^{\rho\gamma}, Q^{\rho\gamma}][A],$$

*N,M* are known and they are the same components of forces and moment present in classical shell theories in composite materials.

$$N = \sum_{k=1}^{n} \int_{h_{k-1}}^{h_k} [\sigma]^k dz, \ M = \sum_{k=1}^{n} \int_{h_{k-1}}^{h_k} [\sigma]^k dz, \ M^* = \sum_{k=1}^{n} \int_{h_{k-1}}^{h_k} [\sigma]^k \, dz \qquad (2.29)$$

$$N = \sum_{k=1}^{n} \int_{h_{k-1}}^{h_k} [\sigma]^k \, dz$$

$$= \sum_{k=1}^{n} \int_{h_{k-1}}^{h_k} [Q]_k dz \begin{bmatrix} e_{xx} \\ e_{yy} \\ 0 \\ 0 \\ 0 \\ e_{xy} \end{bmatrix} - \sum_{k=1}^{n} \int_{h_{k-1}}^{h_k} [Q]_k z dz \begin{bmatrix} K_{xx} \\ K_{yy} \\ 0 \\ 0 \\ 0 \\ K_{xy} \end{bmatrix} + \sum_{k=1}^{n} \int_{h_{k-1}}^{h_k} [Q]_k z^2 dz \begin{bmatrix} Q_{xx} \\ Q_{yy} \\ 0 \\ 0 \\ 0 \\ Q_{xy} \end{bmatrix}. \qquad (2.30)$$

$$M = \sum_{k=1n}^{n} \int_{h_{k-1}}^{h_k} [\sigma]^k dz = \sum_{k=1}^{n} \int_{h_{k-1}}^{h_k} [Q]_k z dz \begin{bmatrix} e_{xx} \\ e_{yy} \\ 0 \\ 0 \\ 0 \\ e_{xy} \end{bmatrix} - \sum_{k=1}^{n} \int_{h_{k-1}}^{h_k} [Q]_k z^2 dz \begin{bmatrix} K_{xx} \\ K_{yy} \\ 0 \\ 0 \\ 0 \\ K_{xy} \end{bmatrix}$$

$$+ \sum_{k=1}^{n} \int_{h_{k-1}}^{h_k} [Q]_k z^3 dz \begin{bmatrix} Q_{xx} \\ Q_{yy} \\ 0 \\ 0 \\ 0 \\ Q_{xy} \end{bmatrix}. \qquad (2.31)$$

While $M^*$ is inner moment component due to Gauss curvature derived from the third fundamental form scarcely used in literature.

$$M^* = \sum_{k=1}^{n} \int_{h_{k-1}}^{h_k} [\sigma]^k dz = \sum_{k=1}^{n} \int_{h_{k-1}}^{h_k} [Q]_k z^2 dz \begin{bmatrix} e_{xx} \\ e_{yy} \\ 0 \\ 0 \\ 0 \\ e_{xy} \end{bmatrix} - \sum_{k=1}^{n} \int_{h_{k-1}}^{h_k} [Q]_k z^3 dz \begin{bmatrix} K_{xx} \\ K_{yy} \\ 0 \\ 0 \\ 0 \\ K_{xy} \end{bmatrix} +$$

$$\sum_{k=1}^{n} \int_{h_{k-1}}^{h_k} [Q]_k z^4 dz \begin{bmatrix} Q_{xx} \\ Q_{yy} \\ 0 \\ 0 \\ 0 \\ Q_{xy} \end{bmatrix}. \tag{2.32}$$

$$\begin{bmatrix} N_{xx} \\ N_{yy} \\ 0 \\ 0 \\ 0 \\ N_{xy} \end{bmatrix} = A \begin{bmatrix} e_{xx} \\ e_{yy} \\ 0 \\ 0 \\ 0 \\ e_{xy} \end{bmatrix} - B \begin{bmatrix} K_{xx} \\ K_{yy} \\ 0 \\ 0 \\ 0 \\ K_{xy} \end{bmatrix} + C \begin{bmatrix} Q_{xx} \\ Q_{yy} \\ 0 \\ 0 \\ 0 \\ Q_{xy} \end{bmatrix}. \tag{2.33}$$

$$\begin{bmatrix} M_{xx}^* \\ M_{yy}^* \\ 0 \\ 0 \\ 0 \\ M_{xy}^* \end{bmatrix} = -B \begin{bmatrix} e_{xx} \\ e_{yy} \\ 0 \\ 0 \\ 0 \\ e_{xy} \end{bmatrix} + C \begin{bmatrix} K_{xx} \\ K_{yy} \\ 0 \\ 0 \\ 0 \\ K_{xy} \end{bmatrix} - D \begin{bmatrix} Q_{xx} \\ Q_{yy} \\ 0 \\ 0 \\ 0 \\ Q_{xy} \end{bmatrix}. \tag{2.34}$$

$$\begin{bmatrix} M_{xx} \\ M_{yy} \\ 0 \\ 0 \\ 0 \\ M_{xy} \end{bmatrix} = C \begin{bmatrix} e_{xx} \\ e_{yy} \\ 0 \\ 0 \\ 0 \\ e_{xy} \end{bmatrix} - D \begin{bmatrix} K_{xx} \\ K_{yy} \\ 0 \\ 0 \\ 0 \\ K_{xy} \end{bmatrix} + E \begin{bmatrix} Q_{xx} \\ Q_{yy} \\ 0 \\ 0 \\ 0 \\ Q_{xy} \end{bmatrix}. \tag{2.35}$$

The derived macroscopic behavior law of a layered composite material can be writen as below in Eq (2.36):

$$\begin{bmatrix} N \\ M \\ M^* \end{bmatrix} = A \begin{bmatrix} e \\ K \\ Q \end{bmatrix} = \begin{pmatrix} A & -B & C \\ -B & C & -D \\ C & -D & E \end{pmatrix} \begin{bmatrix} e \\ K \\ Q \end{bmatrix}, \tag{2.36}$$

where $A = (A_{ij})_{1 \le i,j \le 6} = \sum_{k=1}^{n} [Q]_k (h_k - h_{k-1})$, $\tag{2.37}$

$$B = (B_{ij})_{1 \le i,j \le 6} = \frac{1}{2}\sum_{k=1}^{n}[Q]_k(h_k^2 - h_{k-1}^2),  \qquad (2.38)$$

$$C = (C_{ij})_{1 \le i,j \le 6} = \frac{1}{3}\sum_{k=1}^{n}[Q]_k(h_k^3 - h_{k-1}^3),  \qquad (2.39)$$

$$D = (D_{ij})_{1 \le i,j \le 6} = \frac{1}{4}\sum_{k=1}^{n}[Q]_k(h_k^4 - h_{k-1}^4),  \qquad (2.40)$$

$$E = (E_{ij})_{1 \le i,j \le 6} = \frac{1}{5}\sum_{k=1}^{n}[Q]_k(h_k^5 - h_{k-1}^5),  \qquad (2.41)$$

we then deduce that:

$$a(U, V) = \int_S (e, K, Q)[A]\begin{bmatrix} e \\ K \\ Q \end{bmatrix} = \int_\Omega f.V \, d\Omega + \int_{\partial\Omega} g.V \, dS = L(V),  \qquad (2.42)$$

$[A]$ is the matrix in Eq (2.28).

- The discrete problem on a layer-k

The static equilibrium equations of a ply-k deducted from Eq (2.42) reads:

$$A_1(u, v)|_k = L(v)|_k;  \qquad (2.43)$$

the following matrix equation holds

$$A_{1k}(\hat{U}, \hat{V}) = \hat{V}^T[K_{Gk}^{N-T}]\hat{U};  \qquad (2.44)$$

$$A_{1k}(u, v) = \frac{1}{2}\int_{\Omega k}(\boldsymbol{\varepsilon}_m^t Q_k \boldsymbol{\varepsilon}_m - z\boldsymbol{\varepsilon}_m^t Q_k \boldsymbol{\varepsilon}_b + z^2\boldsymbol{\varepsilon}_m^t Q_k \boldsymbol{\varepsilon}_g - z\boldsymbol{\varepsilon}_b^t Q_k \boldsymbol{\varepsilon}_m + z^2\boldsymbol{\varepsilon}_b^t Q_k \boldsymbol{\varepsilon}_b - z^3\boldsymbol{\varepsilon}_b^t Q_k \boldsymbol{\varepsilon}_g$$
$$+ z^2\boldsymbol{\varepsilon}_g^t Q_k \boldsymbol{\varepsilon}_m - z^3\boldsymbol{\varepsilon}_g^t Q_k \boldsymbol{\varepsilon}_b + z^4\boldsymbol{\varepsilon}_g^t Q_k \boldsymbol{\varepsilon}_g)d\Omega = L(v)  \qquad (2.45)$$

The components of deformation tensor are expressed as a product of the matrix of interpolation functions' gradient and that's of the total displacement vector over the triangular element. From Eqs (2.36–2.42), the stress and strain tensor components are deduced in function of the product above.

$$\boldsymbol{\varepsilon}_m = e_{\alpha\beta} = D_m A = D_m P^{-1}\hat{U} = B_m \hat{U} \quad \text{Where } B_m = D_m P^{-1}  \qquad (2.45A)$$

$$\sigma_m = c\boldsymbol{\varepsilon}_m = cB_m \hat{U}  \qquad (2.45B)$$

Also, the bending stress and strain are expressed

$$\boldsymbol{\varepsilon}_b = k_{\alpha\beta} = D_b A = D_b P^{-1}\hat{U} = B_b \hat{U} \text{ where } B_b = D_b P^{-1};  \qquad (2.45C)$$

$$\sigma_b = c\boldsymbol{\varepsilon}_b = cB_b \hat{U}.  \qquad (2.45D)$$

Finally, as previous,

$$\varepsilon_g = Q_{\alpha\beta} = D_g A = D_g P^{-1} \hat{U} = B_g \hat{U} \quad \text{where } B_g = D_g P^{-1}; \tag{2.45E}$$

$$\sigma_g = c\varepsilon_g = cB_g \hat{U}. \tag{2.45F}$$

The right-hand side of the best first order two-dimensional variational equation reads

$$L(\hat{V})_k = (\int_{S_k} p^i \eta_i dS + \int_{\partial S_k} \bar{p}^i \eta_i d\gamma) = \hat{V}^T \sum_{n=1}^{N_S} (\int_{S_{ek}} N^T p^i dS_{ek} + \int_{\partial S_{ek}} N^T \bar{p}^i d\gamma)$$

$$L(\hat{V})_k = \hat{V}^T (F_v + M)_k \tag{2.46}$$

Here $p^i$ and $\bar{p}^i$ are defined as in R. Nzengwa et al. [25]. Let $F_{Gk}$ be the resultant force vector from distributed load $F_V + M$ such that:

$$F_v = \int_{S_e} N^T p^i dS; \tag{2.47}$$

$$M = \int_{\partial S_e} N^T \bar{p}^i d\gamma. \tag{2.48}$$

By equating $A_{1k}(u, v)$ to $L(\hat{V})_k$, we obtain the structural discrete equilibrium equation over the whole area of *layer-k* as follows:

$$K_{Gk}^{N-T} \hat{U} = F_{Gk}. \tag{2.49}$$

Over a single curved element of area $s_{ek}$ the elementary stiffness matrix $K_{ek}$ and elementary force vectors $F_{ek}$ are locally calculated as follow:

$$K_{ek}^{N-T} = \int_{S_{ek}} (B_m^t A B_m - B_m^t B B_b + B_m^t C B_g - B_b^t B B_m + B_b^t C B_b - B_b^t D B_g + B_g^t C B_m - B_g^t D B_b$$

$$+ B_g^t E B_g) dS_{ek} \tag{2.50}$$

$$F_{ek} = F_v + M + P_e^i. \tag{2.51}$$

$P_e^i$ denotes element nodal concentrated load.

By regular assembling process of elementary stiffness matrix $K_{ek}$ and elementary force vectors $F_{ek}$, we then express the global stiffness matrix $K_{Gk}^{N-T}$ and force $F_{Gk}$ vector over the whole *layer-k* as follow:

$$K_{Gk}^{N-T} = \sum_{n=1}^{N_{Sk}} K_{ek}; \tag{2.52}$$

$$F_{Gk} = \sum_{n=1}^{N_{Sk}} F_{ek}. \tag{2.53}$$

Where $N_{Sk}$ is the total number of elements of the discretized *layer-k*.

• The discrete problem on a laminate

Eqs (2.27) to (2.42) guide us on the homogenization process applied to mono-layer laminate composites material approach. By integrating and summing through the thickness-z each *layer-k* stiffness matrix $K_{Gk}^{N-T}$ and global layer force vectors $F_{Gk}$, we then express the global stiffness matrix $K_{GC}^{N-T}$ and force $F_{GC}$ vector over the whole *layer-k*.

## 3. Results and discussions

This section stands for the investigation of the performance and the accuracy of CSFE-sh elements on composite shell structures behavior through standard tests of laminate shells. The survey analyses the static aspect according to the formulations detailed above. This element is precisely to simulate the behavior of curved structures, namely cylindrical, spherical shells, subjected to axisymmetric and asymmetric loads. The purpose here is to investigate accuracy of the method by increasing the number of elements and to verify the capacity of these elements to simulate behaviors which are considered as problems complex in engineering.

### 3.1. Algorithm for calculating a laminate by the finite element method

The finite element method is used to predict the behavior of the laminate following a known axisymmetric and asymmetric loading. The procedure for implementing our problem numerically is described as follows:

a. Entries of the geometric parameters of the shell.

b. Input of the characteristics and mechanical properties of each ply (direction of the fibers, thickness, etc.).

c. number of layers and thickness of each layer.

d. Parameterization of the shell mesh.

e. Display of the tables of nodes and connectivity and of the coordinates of the mesh.

f. Force loading data entries.

g. Computation of the matrix $[Q]_k$ for each layer; in the local and global reference.

h. Computation of the stiffness of the homogenized composite for the study of the overall behavior of the laminate: matrix A of Eq (2.28).

i. Input of shape functions of the CSFE-sh element and construction of the matrix of shape functions of $e_{\alpha\beta}$, $k_{\alpha\beta}$, $Q_{\alpha\beta}$, respectively $B_m$, $B_k$, $B_Q$ of Eqs (2.50–2.51).

j. Calculation of the stiffness matrices $K_{ek}^{N-T}$ and $F_{ek}$ of Eqs (2.50–2.51) on each element of the mesh, and finally calculation by assembly of the overall stiffness and force matrix ($K_{GC}^{N-T}$ and $F_{GC}$) on a layer-k.

k. solves the system of Eqs (2.52–2.53).

### 3.2. Applications to laminated composite shell structures

• Composite shells under axisymmetric loading.

Benchmark 1: Cylindrical laminated shell under uniform internal pressure.

The test consists of studying the deformation of a composite shell under internal pressure [27]. The one-layer orthotropic shell ($0^0$) as well as a two-layer cross-ply laminate shell ($0^0/90^0$)

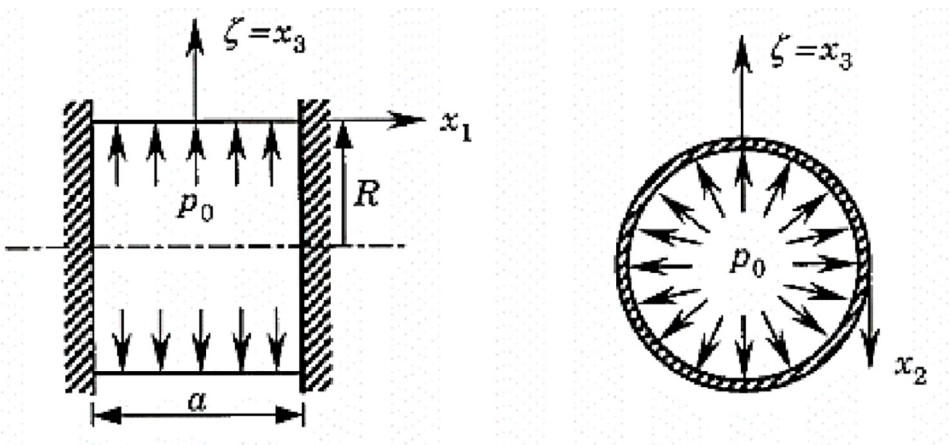

**Fig 8. Cylindrical laminate shell with embedded-ends under internal pressure.**

is analyzed. The shell is embedded at both ends (Fig 8). Due to symmetry, just one-eighth of the structure is analyzed.

**Interpretation.** The results obtained by this model are compared with those obtained analytically by Rao [28] and Timochenko [29] as well as those obtained by the numerical model of Reddy [27]. These two results (see Table 1) being obtained by classical theory do not take into account the transverse shear strain. Accurate results of radial deflection were obtained by the present model without the use of any correction factor withing a particular behaviour (Fig 9). The relative error less than 0.90% here can be due to the contribution of the change of metric in the thickness. It should be noted that several models do not take into account the third fundamental form in the theories of shells.

Benchmark 2: Cylindrical composite laminate shell under opposed loads.

In this test, the structure is a pinched cylindrical laminate shell of revolution, supported at its ends by rigid diaphragms and loaded by two diametrically opposed forces which act at the midpoints of the shell (see Fig 10). This benchmark is a difficult test for shell elements given the type of complex deformation where the bending effect dominates structures under concentrated loads. Due to the symmetry of the structure, one-eighth of the cylinder is computed. The geometry and characteristics of the material used are as follows:

✓ The material properties are as follows:

$$E_1 = 25E_2, \; G_{12} = G_{13} = 0.5 \, E_2, \; G_{23} = 0.5E_2, \; v_{12} = 0.25.$$

The study is carried out for different slenderness S of the thickness, with $R/h = S$.

✓ The geometric parameters of the shell are: $L = 600 \, in.$, $R = 300 \, in.$

✓ The boundary conditions: $U = W = 0$ sur le côté AD

**Table 1. Maximum radial deflection ($W_0$ *in.*) of the cylindrical laminate shell with embedded-ends under internal pressure.**

| Designation of laminate | Reddy [27] | | N-T/CSFE3-sh | | | Ref. [28] | Ref. [29] | Error |
|---|---|---|---|---|---|---|---|---|
| | 4×4Q4 | 2×2Q9 | 2×2 | 4×4 | 8×8 | | | % |
| $0^0$ | 0.3754 | 0.3727 | **0.33570** | 0.3485 | 0.372 | 0.366 | 0.367 | **0.90%** |
| $0^0/90^0$ | 0.1870 | 0.1803 | **0.16785** | 0.17425 | 0.1869 | — | — | **0.05%** |

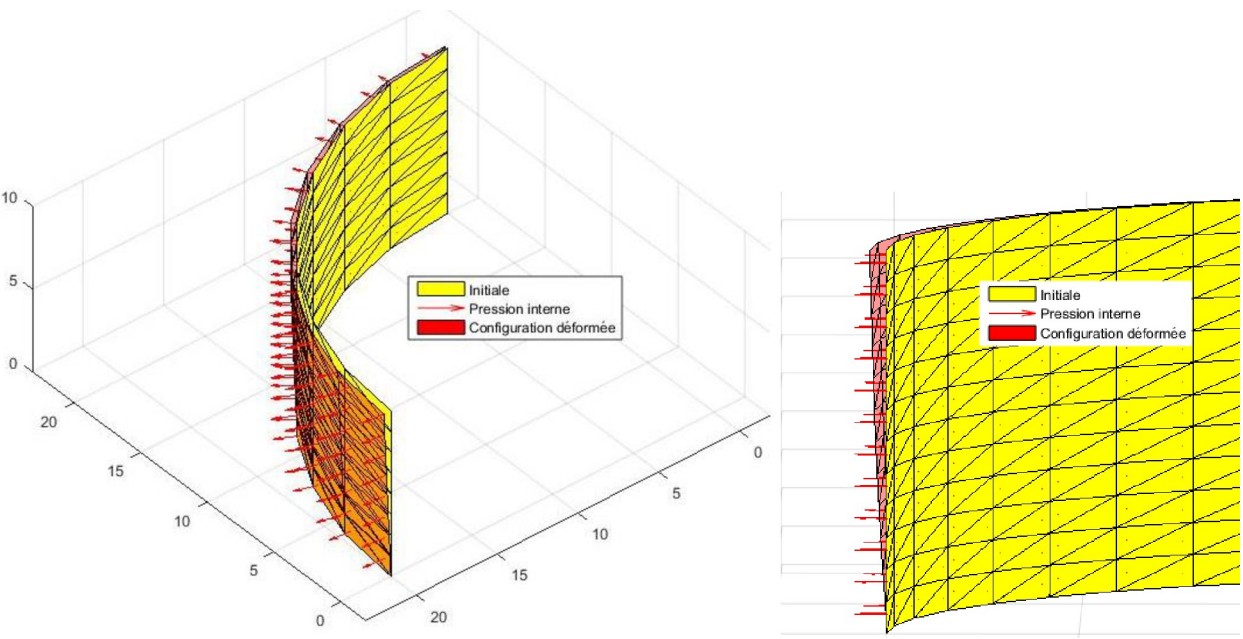

**Fig 9. Isometric view of initial configuration and deformed 1/8 of the cylinder.**

✓ Conditions of symmetry: $U = 0$ on the side CD; $V = 0$ on the side BC; $U = 0$ on the side AB

In Table 2 the values of transverse displacement at point C and tangential displacement at D are presented. The values of the dimensionless displacement at C and D of Reddy are given by:

$$\underline{W_C} = \frac{10h^3 E_L}{PR^2} W; \quad \underline{V_D} = \frac{10h^2 E_L}{PR} V \tag{3.1}$$

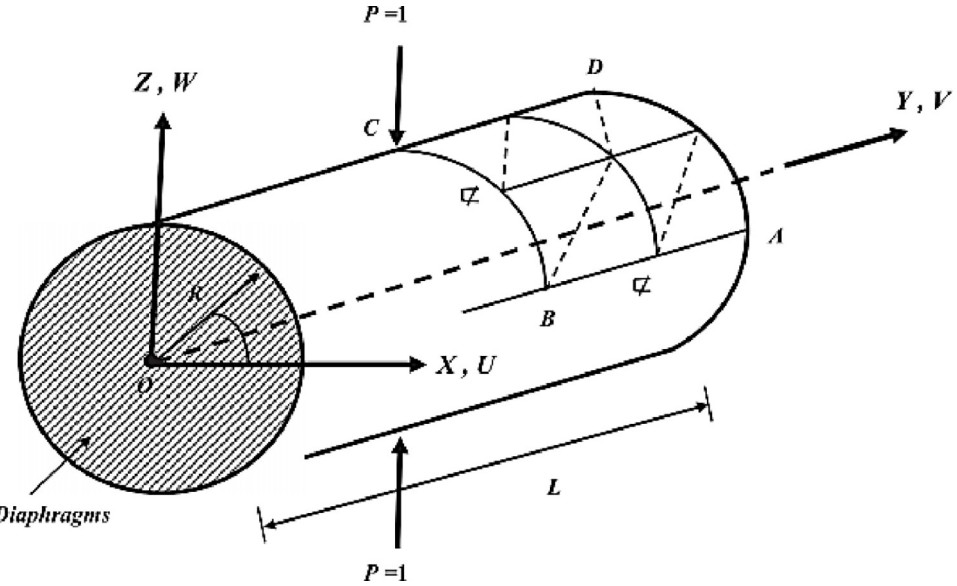

**Fig 10. Pinched cylindrical laminate shell with rigid diaphragm at the ends.**

**Table 2. Radial displacement $W_C$ and tangential displacement $V_D$ of the composite laminate shell clamped with diaphragms, Reddy [9].**

| $\left(\frac{R}{h}\right)$ | Models | Designation of laminates | | | | Error |
|---|---|---|---|---|---|---|
| | | $0^0/90^0$ | | $0^0/90^0/0^0/90^0/0^0)_s$ | | |
| | | $-W_C$ | $-V_D$ | $-W_C$ | $-V_D$ | |
| 50 | NT/CSFE3-sh (4×4) | **1.682** | **1.14** | **0.4504** | **0.436** | 29.33% |
| | NT/CSFE3-sh (8×8) | **2.140** | **1.60** | **0.6250** | **0.537** | 09.97% |
| | NT/CSFE3-sh (16×16) | **2.440** | **1.80** | **1.5900** | **0.570** | -02.52% |
| | Reddy [27] | 2.37756 | 1.4257 | 1.4527 | 0.7383 | |
| 100 | NT/CSFE3-sh (4×4) | **0.600** | **0.765** | **0.3786** | **0.463** | 31.11% |
| | NT/CSFE3-sh (8×8) | **1.036** | **0.835** | **0.6794** | **0.578** | 14.00% |
| | NT/CSFE3-sh (16×16) | **1.395** | **1.248** | **0.8310** | **0.730** | -08.61% |
| | Reddy [27] | 1.2450 | 1.1712 | 0.7405 | 0.6721 | 00.00% |

**Interpretation.** The results obtained by this model are compared with those obtained by the numerical model of Reddy [27]. Satisfactory results were obtained by this model for displacements. We notice a fairly good convergence of the present finite element model. We notice that $V_D$ converges faster than $W_C$.

Benchmark 3: Crossed multilayer cylinder under sinusoidal loading.

In this test we analyze a cylindrical composite shell laminated with three crossed ($90^0/0^0/90^0$), simply supported under sinusoidal pressure with quad circumferential waves. This benchmark was rather proposed by Ren [30, 31] and revisited by Varadan and Bhaskar [32] where the analytical solution based on 3D elasticity is given.

Due to the symmetry just a quarter of the structure (represented by the area ABCD) will be analyzed as seen in Fig 11 below.

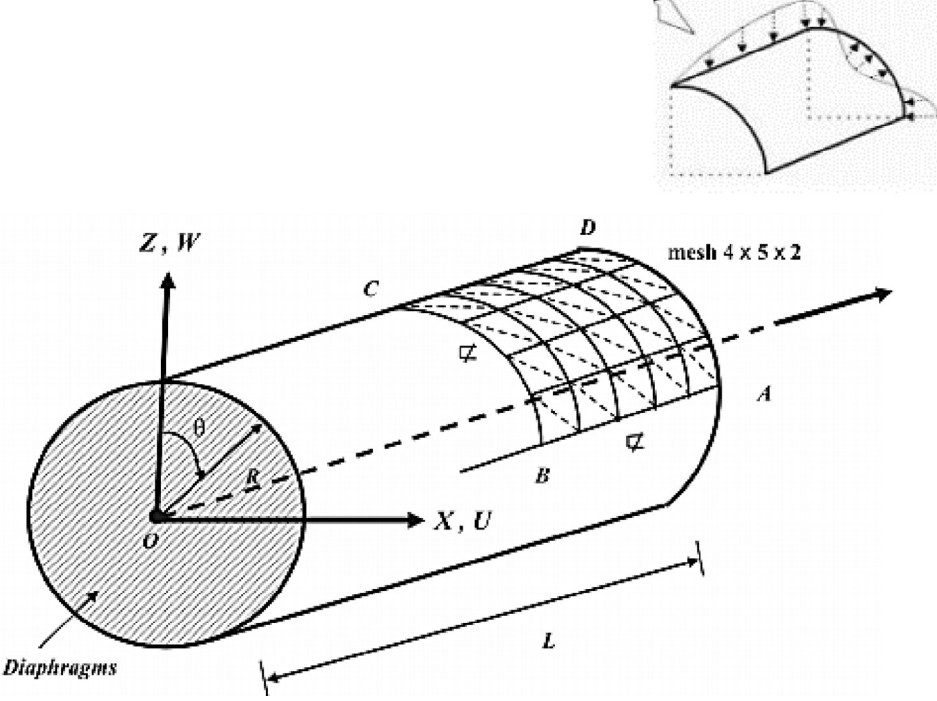

**Fig 11. Cylindrical crossed multi-layer shell simply supported.**

**Table 3. Convergence of deflection $\underline{W}_C$ ($R/h$ = 50.**

| N×M | R/h = 50 | | | | | Error |
|---|---|---|---|---|---|---|
| | DKMT18 | DKMQ24 | MITC4 | CSFE3-sh | | % |
| 4×5 | 0.4739 | 0.441 | 0.3411 | 0.4548 | | 17.23% |
| 8×10 | 0.5229 | 0.5156 | 0.4909 | 0.4846 | | 11.79% |
| 16×20 | 0.5411 | 0.5380 | 0.5319 | 0.514 | 0.520 | 06.46% |
| 32×40 | 0.5446 | 0.5439 | 0.5423 | 0.523 | | 04.82% |
| 64×80 | 0.5467 | 0.5464 | 0.5461 | 0.535 | | 02.64% |
| Ref [30] | 0.5495 | | | | | 00.00% |

**Table 4. Convergence of deflection $\underline{W}_C$ ($R/h$ = 100).**

| N×M | R/h = 100 | | | | Error |
|---|---|---|---|---|---|
| | DKMT18 | DKMQ24 | MITC4 | CSFE-sh | % |
| 4×5 | 0.3873 | 0.3766 | 0.2957 | 0.2981 | 36.77% |
| 8×10 | 0.4468 | 0.4447 | 0.4247 | 0.4321 | 08.35% |
| 16×20 | 0.4653 | 0.4647 | 0.4598 | 0.4487 | 04.83% |
| 32×40 | 0.4704 | 0.4700 | 0.4687 | 0.4656 | 01.25% |
| 64×80 | 0.4712 | 0.4714 | 0.4611 | 0.4713 | 00.04% |
| Ref [31] | 0.4715 | | | | 00.00% |

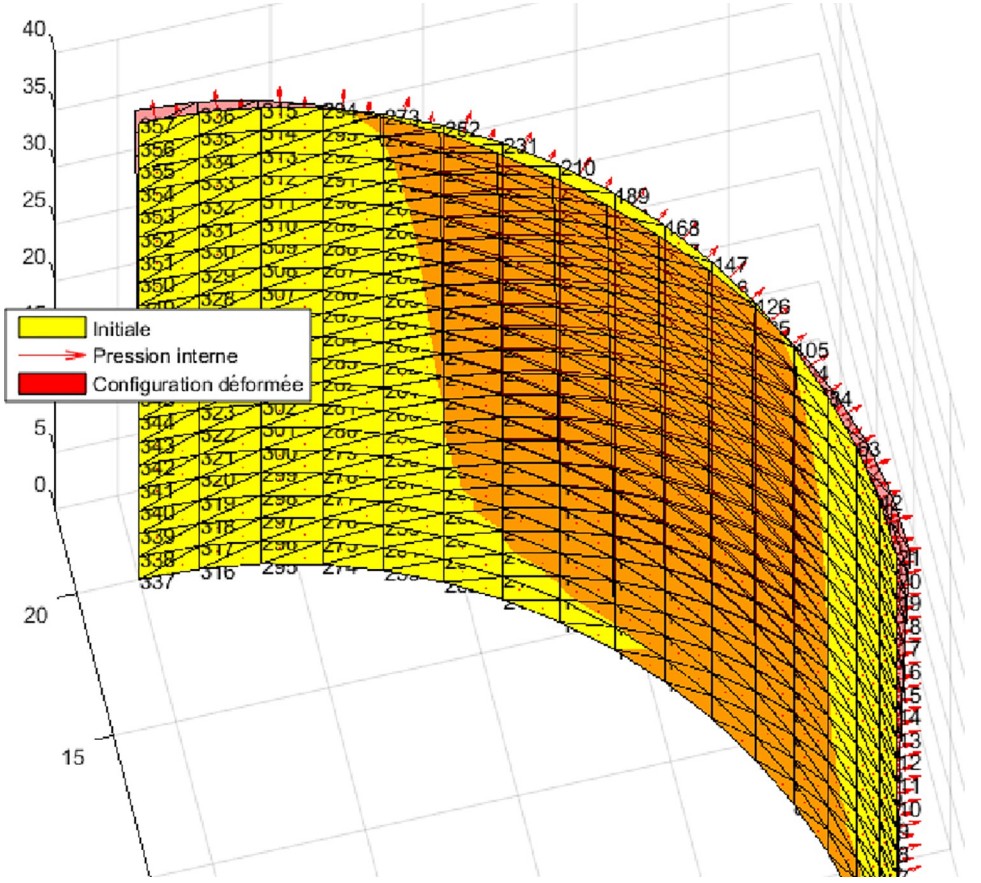

**Fig 12. Isometric view of the deformed configuration of 1/8 of the cylinder for a 6×20 mesh.**

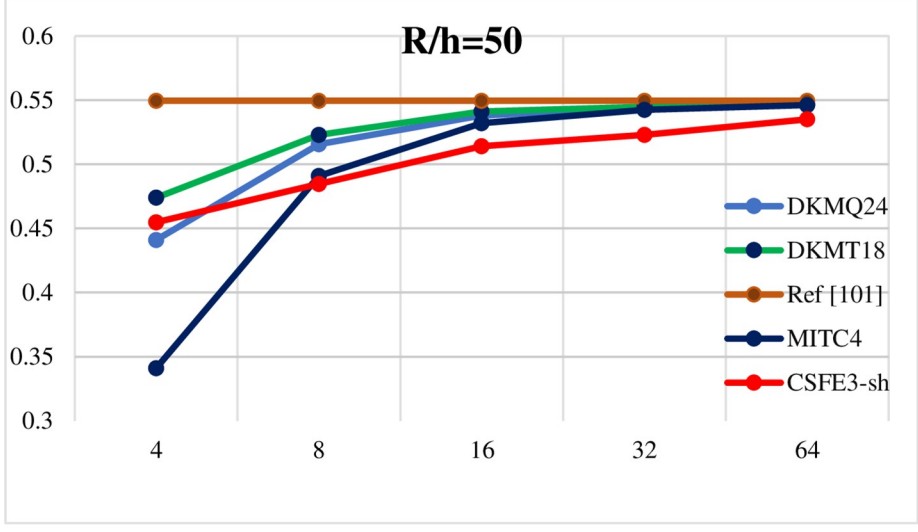

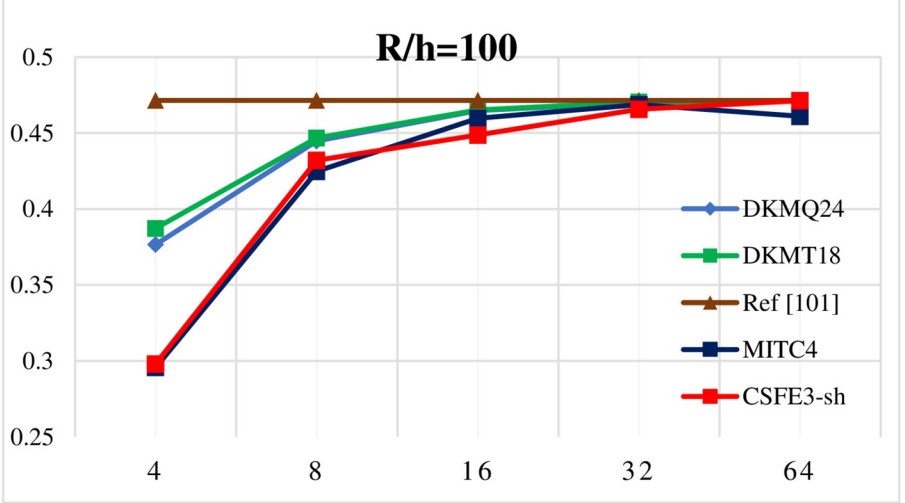

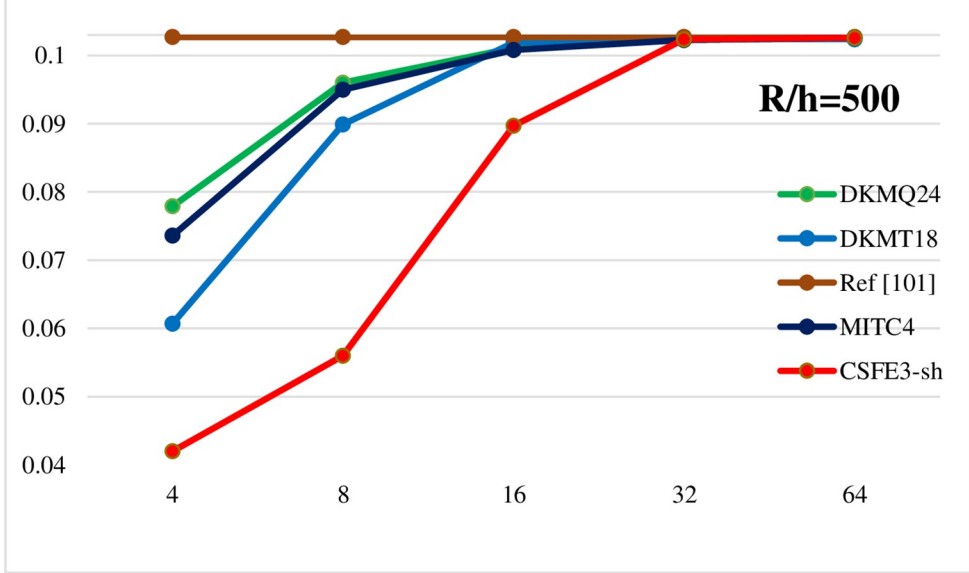

**Fig 13. Charts of defection convergence at point C (R/h = 50, R/h = 100 et R/h = 500).**

**Table 5. Convergence of deflection $\underline{W}_C$ ($R/h = 500$).**

| N×M | R/h = 500 | | | | Error |
|---|---|---|---|---|---|
| | DKMT18 | % | MITC4 | CSFE-sh | % |
| 4×5 | 0.0607 | 0.0779 | 0.0736 | **0.042** | 59.10% |
| 8×10 | 0.0899 | 0.0960 | 0.0950 | **0.056** | 45.47% |
| 16×20 | 0.1019 | 0.1011 | 0.1008 | **0.0897** | 12.65% |
| 32×40 | 0.1024 | 0.1023 | 0.1023 | **0.1024** | 00.29% |
| 64×80 | 0.1024 | 0.1026 | 0.1026 | **0.1025** | 00.19% |
| Ref [27, 30] | 0.1027 | | | | 00.00% |

The dimensionless properties of the material are:

$$^{E_1}/_{E_2} = 25, \; G_{12} = G_{13} = 0.5E_2, \; G_{23} = 0.2E_2, \; \upsilon_{12} = 0.25.$$

The study is carried out for different slenderness S of the thickness, with $R/h = S$.
The sinusoidal loading is given by: $Q = f_0 \sin\left(\frac{\pi z}{L}\right) \cos(4\theta)$;
The geometric parameters of the shell are: $L = 80m$, $R = 20m$;
The boundary conditions: $U = W = 0$ on the side AD;
Conditions of symmetry: $W = 0$ on side AB; $V = 0$ on the side BC; $U = 0$ on the side CD.
The value of displacement at point C is compared with the reference solution.
The value of reference of the central deflection $W_C$ is given by:

$$\underline{W}_C = \frac{10E_L}{f_0 . S^4 h} W; \; S = \left(\frac{R}{h}\right). \tag{3.2}$$

Tables 2–4 compares elements DKMT18, DKMQ24, MITC4 [33] and CSFE3-sh for various ratios. The discretization is made in **N×M×2** triangular elements.

**Interpretation.** We observe (Fig 12) that for the case $90^0/0^0/90^0$, the model gives quite good results compared to the reference solution. It can be seen that the element CSFE3-sh converges less quickly than the elements MITC 4, DKMT 18 and DKMQ24 (Fig 13) and (Table 5). However, for R / h = 100 the element CSFE3-sh converges faster than the element MITC4. The

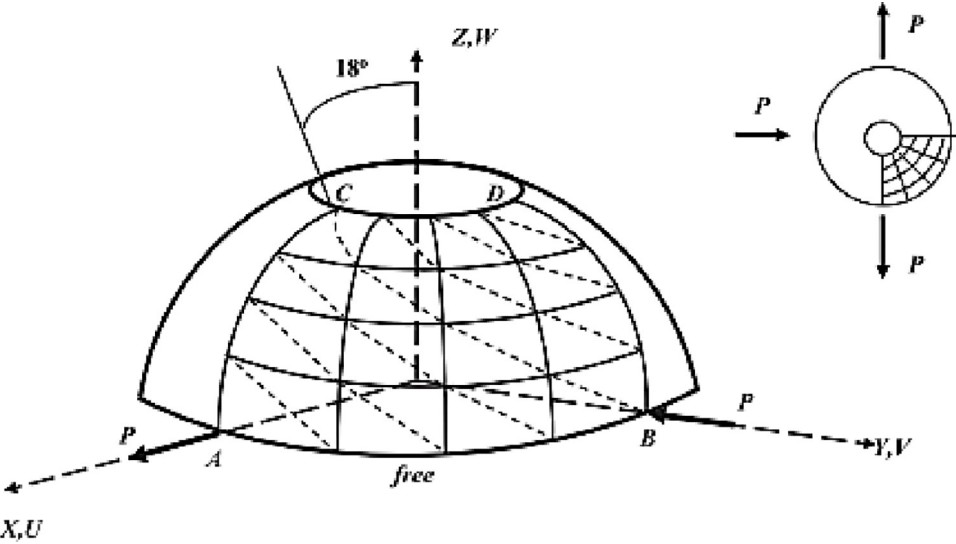

**Fig 14. Spherical shell bi-embedded under uniform internal pressure.**

**Table 6. Maximum deflection ($W_0$) of the spherical composite shell embedded under internal pressure uniformly distributed.**

| $90^0/0^0$ | | | | Error |
|---|---|---|---|---|
| Number of elements N | Alwar and Narasimhan [35] | N×M | CSFE3-sh | % |
| 12 | 0.14901 | 4×4 | **0.1419** | **03.99%** |
| 14 | 0.14741 | 8×8 | **0.238** | **-61.02%** |
| 16 | 0.14765 | 16×16 | **0.1345** | **09.01%** |
| Ref [35] | 0.14782 | | | 00.00% |

element converges better toward the analytical solution for the different meshes when the ratio R / h becomes large. Due to the fact that the structure undergoes a strong gradient of the load and which requires a more refined mesh, a better precision is noticed when the increase in the load is well represented on the circumference. This is expressed by a mesh of 20 × 16 elements, Table 3. No correction coefficient has been used to boost or to penalize a specific behaviour among the components of energy equation. This good accuracy of result is certainly the effect of the gauss curvature contribution into the potential energy of the structure. By applying N-T shell equations on laminates structures, it appears unnecessary to use a correction factor to compute laminates.

Benchmark 4: Bi-embedded spherical composite shell under uniform internal pressure.

The spherical shell shown in the Fig 14 is often used to assess the performance of thin shells. In this test we analyze a spherical composite shell laminated with two crossed layers ($90^0/0^0$), embedded at the upper and lower edges, under uniformly distributed internal pressure. Due to

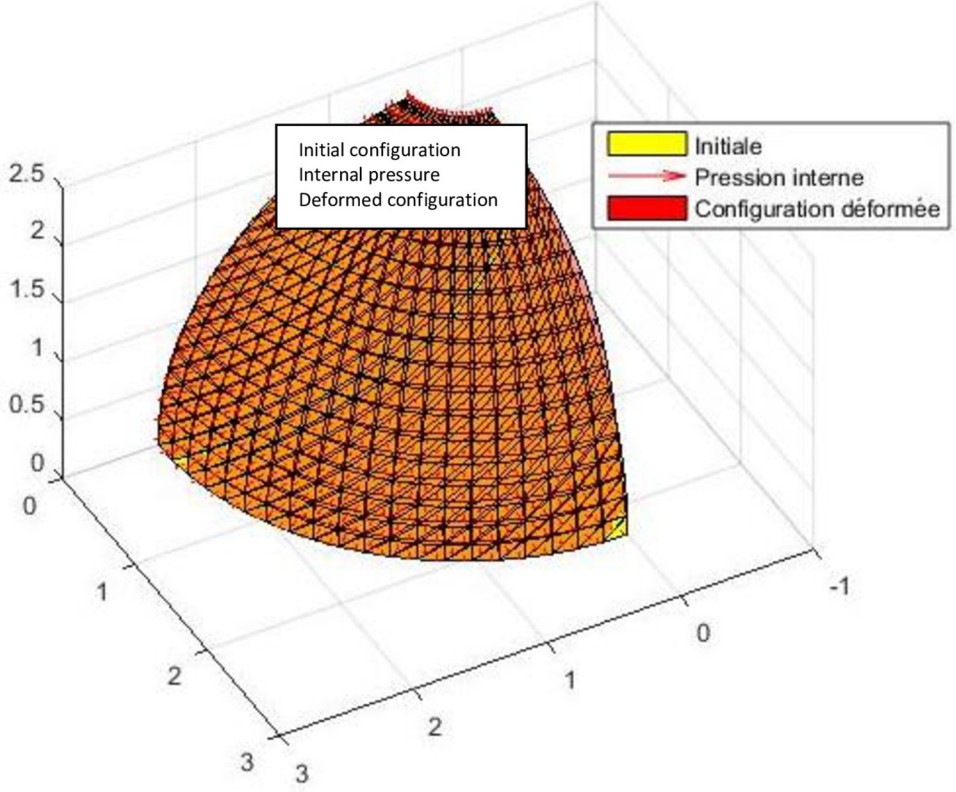

**Fig 15. Isometric view of the deformed configuration of ¼ of the sphere (mesh 16×16).**

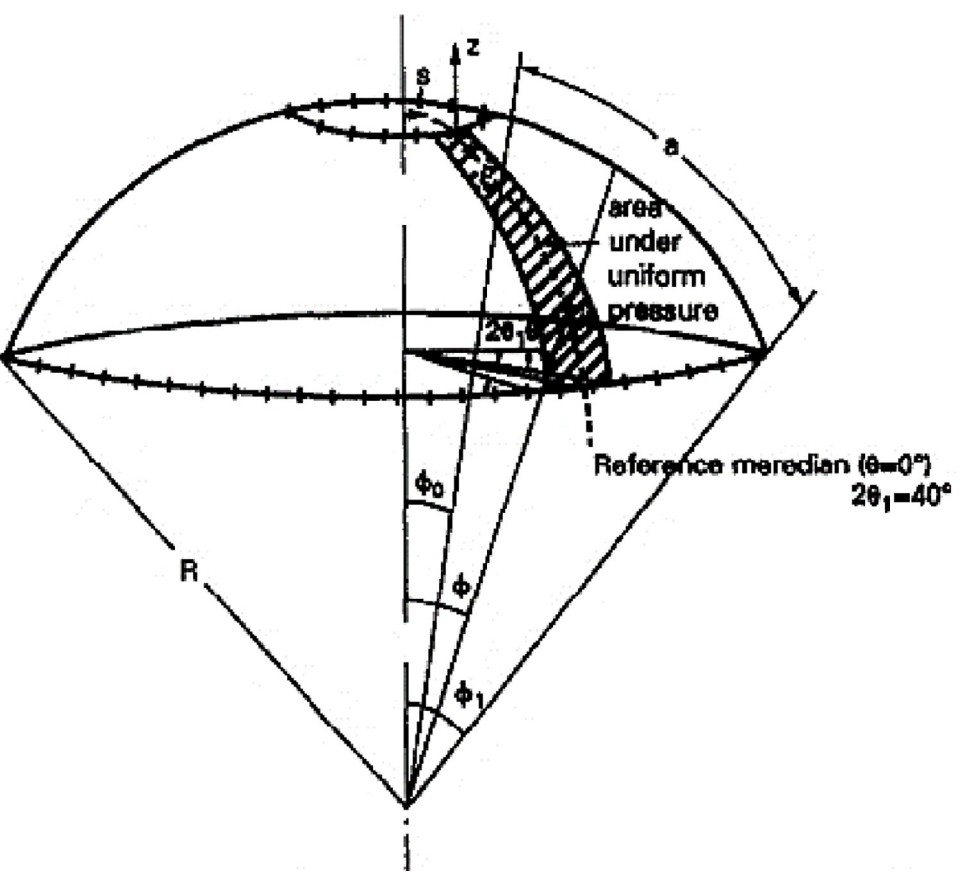

**Fig 16. Spherical shell under asymmetric loading.**

the symmetry just a quarter of the shell (represented by the area ABCD) will be analyzed as seen in the figure below.

The dimensionless properties of the material are:

$$^{E_1}/_{E_2} = 20, \ G_{12} = G_{13} = 0.5E_2, \ G_{23} = 0.2E_2, \ v_{12} = 0.28.$$

The geometric parameters of the shell are: $R/h = 30$, $\varphi = 80$, $\theta = 90$;

The boundary conditions: $U = V = W = 0$ on the sides AB and CD

Conditions of symmetry: $U = 0$ on the side BD and on the side AC;

The maximum value of displacement is compared with the reference solution.

The reference value of the deflection displacement $\underline{W}_C$ is given by [34]:

$$\underline{W}_C = \frac{10^4 E_2 h^3}{qa^4} W. \tag{3.3}$$

**Interpretation.**   The results obtained (Table 6) are compared with the reference solution obtained by Alwar and Narasimhan [35, 36] via the analytical method. One notes an overrun of displacement for a mesh 8 × 8. However, the element exhibits acceptable convergence beyond 16 × 16 (Fig 15).

- **Composite shells under asymmetric loading**

Benchmark 5: Bi-recessed spherical composite shell under asymmetric loading.

In this test we analyze a spherical composite shell laminated with two crossed layers ($90^0$/$0^0$) (Fig 16), embedded at the upper and lower edges, under uniform internal pressure. Due to the symmetry just a quarter of the hull will be analyzed. The geometrical properties and the characteristics of the material and the reference value of the maximum displacement are the same as those above.

The loading is given by: $Q_z = q \cos(\theta)$;

The boundary conditions: $U = V = W = 0$ on sides DC and AB

Conditions of symmetry: $U = 0$ on the side AC and: $U = W = 0$ on the side BD;

**Interpretation.**    The results obtained (Fig 17) are compared with the reference solution obtained by Alwar and Narasimhan [35] via the analytical method. One notices a rather good

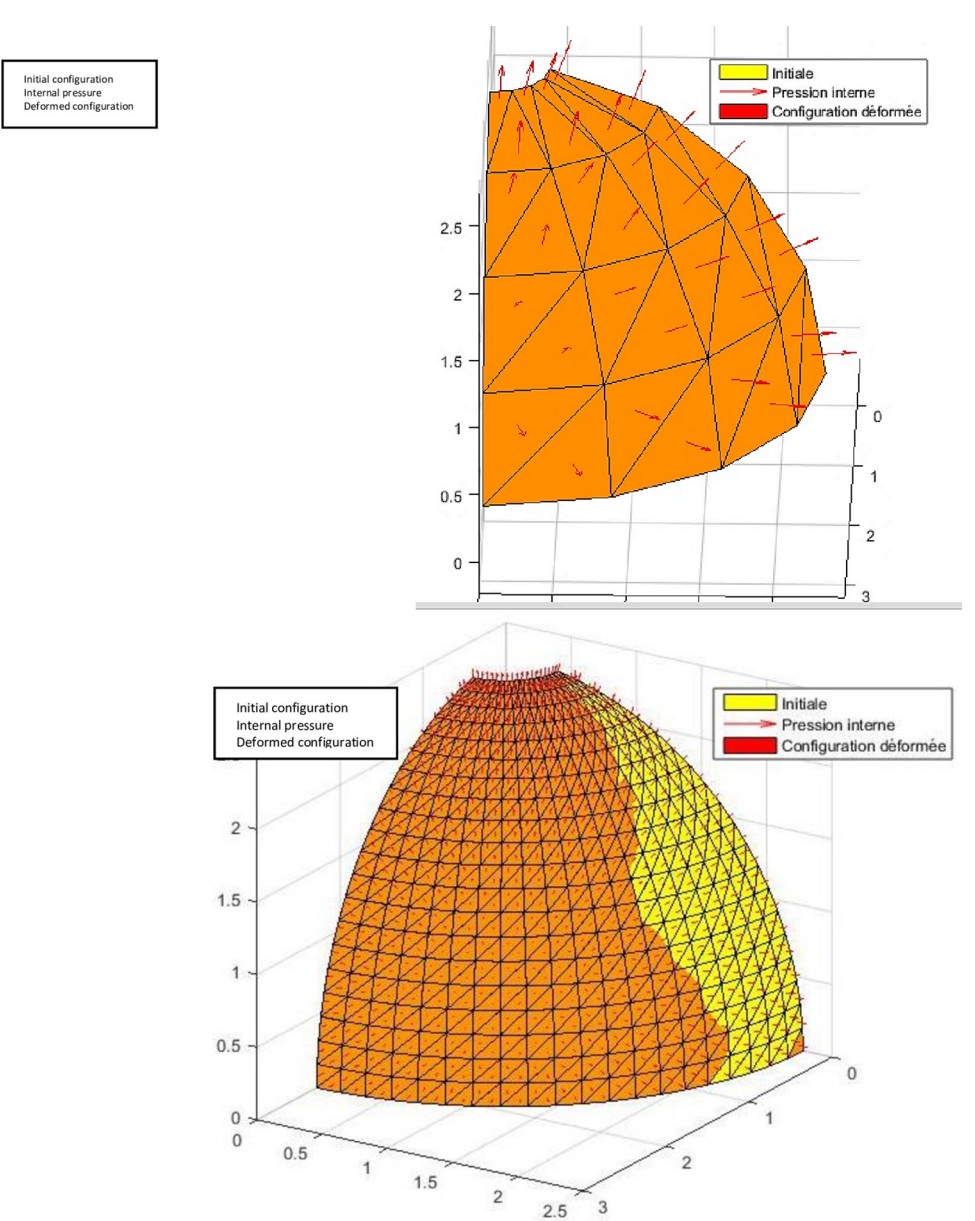

**Fig 17.  Isometric view of the deformed configuration of ¼ of the sphere (meshes 4×4 and 20×20).**

**Table 7. Maximum deflection ($W_0$) of the embedded spherical composite Laminated shell under asymmetric internal pressure.**

| | $(90^0/0^0)$ | | | |
|---|---|---|---|---|
| Number of Elements of N sequence | Alwar et Narasimhan [35] | N×M | NT/CSFE3-sh |
| 12 | 0.50814 | 4×4 | 0.4184 |
| 14 | 0.50788 | 8×8 | 0.4458 |
| 16 | 0.50775 | 16×16 | 0.4614 |

convergence of the deflection towards the reference solution (Table 7). In addition, the maximum deflection is obtained for $\theta = 0^o$.

## 3.3. Conclusion

This framework was about performing the numerical validation of the finite element model (CSFE-sh) based on the N-T theory on layered composite shell structures. For this, we carried out tests using the standards engineering problems found in the literature. We noted that:

- The CSFE-sh finite element model is general because it has been used for any layering scheme, geometry and boundary conditions with accurate results.

- The CSFE3-sh element can be used for both isotropic and composite shells.

- The N-T theory works well without using a correction factor.

- When the R / h ratio increases and therefore the shell becomes thinner, the defection decreases.

- Increasing the number of layers for the same slenderness leads to a decrease in deflection.

When the structure undergoes a strong gradient of the load, this requires a more refined mesh. Better precision is then noticed when the increase in load is well represented on the circumference.

## Author Contributions

**Conceptualization:** Joseph Nkongho Anyi, Alexandra Tchamdjie Pouakam.

**Data curation:** Joseph Nkongho Anyi, Jean Chills Amba.

**Formal analysis:** Joseph Nkongho Anyi, Jean Chills Amba, Merlin Bodol Momha, Robert Nzengwa.

**Investigation:** Joseph Nkongho Anyi, Alexandra Tchamdjie Pouakam, Landry Djopkop.

**Methodology:** Joseph Nkongho Anyi, Jean Chills Amba, Robert Nzengwa.

**Project administration:** Merlin Bodol Momha.

**Resources:** Joseph Nkongho Anyi.

**Software:** Joseph Nkongho Anyi, Fongho Eric, Platon Dongmo Nizegha, Merlin Bodol Momha, Landry Djopkop.

**Supervision:** Jean Chills Amba.

**Validation:** Platon Dongmo Nizegha.

**Visualization:** Fongho Eric, Platon Dongmo Nizegha.

**Writing – original draft:** Joseph Nkongho Anyi, Alexandra Tchamdjie Pouakam.

**Writing – review & editing:** Joseph Nkongho Anyi, Robert Nzengwa.

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
