## [Decision Letter · Decision Letter 0]

11 Apr 2022

INVESTIGATION OF THE MACROSCOPIC BEHAVIOUR OF LAMINATES SHELLS (MBLS) UNDER VARYING LOADS USING LOW ORDER CSFE-sh FEM and the N-T’s 2-D SHELL EQUATIONS.

PONE-D-22-04183

Dear Dr. Joseph,

We’re pleased to inform you that your manuscript has been judged scientifically suitable for publication and will be formally accepted for publication once it meets all outstanding technical requirements.

Kind regards,

Krishna Garikipati, PhD

Academic Editor

PLOS ONE

Additional Editor Comments (optional):

We have received one review of your manuscript. I have made my recommendation to accept on its basis.

Reviewers' comments:

Reviewer's Responses to Questions

**Comments to the Author**

1. Is the manuscript technically sound, and do the data support the conclusions?

Reviewer #1: Yes

2. Has the statistical analysis been performed appropriately and rigorously? 

Reviewer #1: Yes

3. Have the authors made all data underlying the findings in their manuscript fully available?

Reviewer #1: Yes

4. Is the manuscript presented in an intelligible fashion and written in standard English?

Reviewer #1: Yes

5. Review Comments to the Author

Reviewer #1: The paper presents an investigation of the macroscopic behaviour of laminates shells under varying loads using low order CSFE-sh FEM and the N-T’s 2D shell equations. According to the reviewer’s opinion, the paper is well-structured and clear. The topic is interesting and falls within the aim of the journal. In addition, the results are well-presented and could be helpful to further develop the same topic. Therefore, the paper can be accepted for publication in the current form.

6. PLOS authors have the option to publish the peer review history of their article (what does this mean?). If published, this will include your full peer review and any attached files.

Reviewer #1: No

---

## [Editor Report · Acceptance letter]

21 Apr 2022

PONE-D-22-04183 

Investigation of the macroscopic behaviour of laminates shells (MBLS) under varying loads using low order CSFE-sh FEM and the N-T’s 2-D shell equations. 

Dear Dr. Anyi:

I'm pleased to inform you that your manuscript has been deemed suitable for publication in PLOS ONE. Congratulations! Your manuscript is now with our production department. 

Kind regards, 

on behalf of

Prof. Krishna Garikipati 

Academic Editor

PLOS ONE